# Update in Pathogenesis, Diagnosis, and Therapy of Prolactinoma

**DOI:** 10.3390/cancers14153604

**Published:** 2022-07-24

**Authors:** Noriaki Fukuhara, Mitsuru Nishiyama, Yasumasa Iwasaki

**Affiliations:** 1Department of Hypothalamic and Pituitary Surgery, 2-2-2, Toranomon, Minato-ku, Tokyo 105-8470, Japan; 2Okinaka Memorial Institute for Medical Research, 2-2-2, Toranomon, Minato-ku, Tokyo 105-8470, Japan; 3Department of Endocrinology, Metabolism and Nephrology, Kochi Medical School, Kochi University, 185-1 Kohasu, Oko-cho, Nankoku City 783-8505, Japan; 326nishiyama@gmail.com; 4Health Care Center, Kochi University, 2-5-1, Akebono-cho, Kochi 780-8520, Japan; 5Department of Clinical Nutrition, Faculty of Health Science, Suzuka University of Medical Science, 1001-1, Kishioka, Suzuka City 510-0293, Japan; iwasakiyasumasa@gmail.com

**Keywords:** prolactinoma, dopamine agonist, pituitary carcinoma, aggressive pituitary neuroendocrine tumors, SF3B1 mutation

## Abstract

**Simple Summary:**

This review updates recent advances in the pathogenesis, diagnosis, and therapy of prolactinoma. Prolactinomas, comprising 30–50% of all pituitary neuroendocrine tumors, frequently occur in females aged 20 to 50 and cause hypogonadism and infertility. In typical cases, female patients exhibit galactorrhea and amenorrhea due to serum prolactin (PRL) elevation, and during pregnancy, they should be carefully treated. During diagnosis, other causes of hyperprolactinemia must be excluded, and an MRI is useful for detecting pituitary neuroendocrine tumors. For the treatment of prolactinoma, dopamine agonists are effective in decreasing PRL levels and shrinking tumor size in most patients. Surgical treatment is recommended for patients who are resistant or intolerant to dopamine agonists. This review also discusses giant and malignant prolactinomas, prolactinoma-associated pregnancy, and new therapeutic approaches.

**Abstract:**

Prolactinomas comprise 30–50% of all pituitary neuroendocrine tumors, frequently occur in females aged 20 to 50, and cause hypogonadism and infertility. In typical cases, female patients exhibit galactorrhea and amenorrhea due to serum prolactin (PRL) elevation, and patients during pregnancy should be carefully treated. During diagnosis, other causes of hyperprolactinemia must be excluded, and an MRI is useful for detecting pituitary neuroendocrine tumors. For treating prolactinoma, dopamine agonists (DAs) are effective for decreasing PRL levels and shrinking tumor size in most patients. Some DA-resistant cases and the molecular mechanisms of resistance to a DA are partially clarified. The side effects of a DA include cardiac valve alterations and impulse control disorders. Although surgical therapies are invasive, recent analysis shows that long-term remission rates are higher than from medical therapies. The treatments for giant or malignant prolactinomas are challenging, and the combination of medication, surgery, and radiation therapy should be considered. Regarding pathogenesis, somatic SF3B1 mutations were recently identified even though molecular mechanisms in most cases of prolactinoma have not been elucidated. To understand the pathogenesis of prolactinomas, the development of new therapeutic approaches for treatment-resistant patients is expected. This review updates the recent advances in understanding the pathogenesis, diagnosis, and therapy of prolactinoma.

## 1. Introduction

Prolactinomas are common pituitary neuroendocrine tumors (PitNET) derived from prolactin (PRL)-producing cells that cause hypogonadism and infertility due to hyperprolactinemia. It most frequently occurs in young women and is often associated with galactorrhea and amenorrhea. Most cases are sporadic, but prolactinomas may also occur because of mutations of the multiple endocrine neoplasia type 1 (*MEN1*) or aryl hydrocarbon receptor-interacting protein (*AIP*) gene. Medical therapy with a dopamine agonist (DA) is highly effective in the majority of cases [1,2].

## 2. Epidemiology

Prolactinoma accounts for approximately 50% of all pituitary tumors and is the most frequent of all functional PitNETs [2,3]. The annual incidence of prolactinomas is about 2.2 per 100,000 in Finland [4]. It is most common in women of reproductive age, with an incidence of about 10 per 100,000. Peak incidence occurs around 30 years old [5] but is also seen after menopause. The annual incidence of DA-treated hyperprolactinemia is 24 per 100,000 women aged 25 to 34 in the Netherlands [6]. Male patients are relatively rare, and the disease takes longer to be diagnosed because of the lack of clinical symptoms.

## 3. Pathogenesis

Prolactinomas are divided into two subgroups: familial and sporadic. Among the familial cases, the well-known causative mutation is that of *MEN1*. Indeed, prolactinoma is most frequent in familial PitNETs in patients with MEN1 [7]. Although relatively rare, *MEN1* gene mutation is found in sporadic prolactinomas. In addition, prolactinoma is reported to occur in families with a mutation in the *PRKAR1A* (Carney complex), *CDKN1B* (MEN4), or *AIP* (FIPA) gene [8].

The precise molecular mechanism whereby *MEN1* gene mutation causes prolactinoma is still obscure. In endocrine cells, MENIN, a protein product of the *MEN1* gene, has a negative effect on cell growth via the induction of cell cycle-inhibiting genes such as *CDKN1B* (p27^KIP1^) or *CDKN2C* (p18^IKN4C^). Thus, MEN1 is recognized as a tumor-suppressor gene. In this sense, a loss-of-function mutation in one allele of genomic DNA can enhance the cell cycle, followed by the tumorous growth of the affected cells. The reason that the germ line *MEN1* gene mutation causes a PRL-, growth hormone (GH)- or thyroid stimulating hormone (TSH)-producing PitNET as opposed to any other pituitary tumor is not known. One possibility is that MENIN somehow interacts with PIT1 (POU1F1), the common transcription factor specifically expressed in the somato-, mammo-, and thyrotrope cells, and facilitates PIT1-dependent gene transcription that is related to hormone synthesis and cell growth.

The *PRKAR1A* gene encodes the regulatory subunit of protein kinase A (PKA), and a germ line loss-of-function mutation can cause constitutive activation of the catalytic subunit of PKA with a resultant increase in PKA-dependent cell growth and the production of pituitary hormones such as PRL, GH, or TSH. However, the impact of PKA activation on cell proliferation is moderate, and the affected cells are reported to show the phenotype of non-invasive hypertrophy rather than an aggressive tumor.

Regarding sporadic pituitary tumors, recent advances in genome-wide sequencing (GWAS) have identified the mutations responsible for tumorigenesis. Indeed, *USP8* and *gsp* gene mutations have been found in the substantial parts of corticotrope and somatotroph tumors, respectively. Until recently, however, no driver-gene mutation had been reported for prolactinoma, partly because most patients with the tumor receive medical, not surgical, therapy. Recently, a GWAS analysis of somatotroph tumor cells showed aberrant splicing of various mRNAs, including that of the estrogen-related receptor γ (ERRγ) gene (*ESRRG*) [9]. Eventually, Li et al. identified a missense mutation of the splicing factor 3B1 gene (*SF3B1*), causing the amino acid substitution (R625H) [10]. Because the SF3B1 protein is involved in splicing nascent RNA, a variety of abnormally spliced mRNA can result from this mutation. In the case of lactotroph tumor cells, abnormally spliced mRNA derived from the *ESRRG* gene, which encodes ERRγ, results in a mutant ERRγ protein, which has shown to have an abnormally high affinity for PIT1 and potently enhance PIT1-dependent PRL gene transcription. Furthermore, the expression of the DLG1 protein is found to be decreased due to the aberrantly spliced DLG1 mRNA [11]. The DLG1 protein is known to have tumor-suppressor properties, and thus its decrease can cause enhanced lactotroph cell growth. Indeed, a prolactinoma-harboring *SF3B1* gene mutation is reported to have invasive properties.

The pathogenesis of prolactinoma without the *SF3B1* mutation is still unknown. One of the major factors promoting lactotroph tumorigenesis is estrogen; indeed, prolactinoma is known to be more frequent in women than in men. In an animal experiment, prolactinoma was reported to develop a lactotroph tumor in some species of rats in which estrogen was chronically administered. Estrogen is known to increase the expression of a variety of growth factors and oncoproteins such as fibroblast growth factors (FGFs), transforming growth factor β (TGF β), and pituitary tumor transforming gene (PTTG), all of which are known to facilitate tumor growth. More recently, it was reported that estrogen causes epigenetic changes in a variety of tissues other than the adenohypophysis [12]. If estrogen changes the expression level of genes such as the dopamine D2 receptor via epigenetic changes to the pituitary cells, the hormone may influence lactotroph cell growth without a gene mutation. The involvement of an epigenetic mechanism in prolactinoma pathogenesis is not well characterized and awaits further research.

## 4. Diagnosis

PRL is secreted from the lactotroph in the anterior pituitary gland. PRL is regulated mainly by hypothalamic dopamine in an inhibitory manner and by thyrotropin-releasing hormone (TRH) and estrogen in a stimulatory manner. Physiologically, PRL has a positive effect on mammary gland development during pregnancy and milk secretion during the postpartum period. Eating, exercise, and sleep increase serum PRL levels, and they are influenced by the menstrual cycle, pregnancy, and female hormone therapy. These factors should be clarified during a medical interview [1,2,13].

Hyperprolactinemia causes symptoms such as galactorrhea and amenorrhea, the former being due to the direct effect of PRL on the mammary glands and the latter due to the suppression of hypothalamic gonadotropin-releasing hormone (GnRH) and pituitary gonadotropin secretion. It is known that hyperprolactinemia inhibits GnRH neurons via kisspeptin neurons [14]. In men, the suppression of the gonadal system also elicits a decreased libido and pubescence, which are often unrecognized. As a result, headache and visual field disturbance associated with macrotumors are frequently the main complaints in men [1,2,13]. In addition, hypogonadism is a common form of hypopituitarism in men with macroprolactinoma, and the entity of hypogonadism recovery after treatment is a relevant medical issue, mainly in younger patients [15].

Serum PRL levels are in a supraphysiological range during pregnancy and the postpartum period. Since basal serum PRL levels are prone to fluctuation, especially in women, multiple measurements should be performed to ensure reproducibility. Known causes of hyperprolactinemia include prolactinoma, acromegaly (growth-hormone-producing PitNET), hypothalamic and pituitary stalk lesions, medications, primary hypothyroidism, and renal failure [16] (Table 1). In cases of severe hyperprolactinemia (>200 ng/mL), the presence of a prolactinoma should be considered, although it can present with any level of PRL elevation [3,17]. In mild PRL elevation, drug-induced hyperprolactinemia should be considered. The causes of drug-induced hyperprolactinemia include anti-ulcer and antiemetic drugs, antihypertensive drugs, psychotropic drugs, and estrogens. To determine whether it is drug-induced or not, the candidate drug could be discontinued or replaced by other drugs to see if PRL levels return to a normal range within 48–96 h [18]. Almost half of GH-producing PitNETs (acromegaly) simultaneously secrete PRL. An organic lesion in the hypothalamus and pituitary stalk, such as tumors, inflammation, and granulomatous and vascular diseases, could cause hyperprolactinemia, and thus, it is necessary to exclude these diseases. As mentioned above, inhibitory regulation by hypothalamic dopamine is the predominant regulatory mechanism of PRL secretion. Mass lesions of the hypothalamus or pituitary stalk cause the dopamine inhibition of PRL to be insufficient, resulting in hyperprolactinemia. In primary hypothyroidism, the release of a negative feedback regulation can lead to increased hypothalamic TRH secretion, resulting in hyperprolactinemia. If no clinical symptoms are observed in patients with hyperprolactinemia, the possibility of macroprolactinemia should be considered. Macroprolactin consists of high molecular weight PRL, which is a complex of PRL and anti-PRL antibodies and exhibits markedly reduced biological efficacies. In this case, serum PRL levels should be re-assayed after anti-PRL antibodies have been removed by polyethylene glycol treatment [19]. If underlying causative diseases cannot be found, idiopathic hyperprolactinemia is diagnosed. Serum PRL levels were stable in most idiopathic cases after long-term follow up [20,21].

Prolactinoma is diagnosed by imaging tests, such as magnetic resonance imaging (MRI), which detects PitNETs. Additionally, hyperprolactinemia can be caused by a variety of other factors, as described above, and thus these have to be excluded. Serum PRL should be measured in suspected cases of prolactinoma: women with irregular menstruation, amenorrhea, infertility, or galactorrhea, and men with decreased libido. After hyperprolactinemia is confirmed, various differential diagnoses should be made, including prolactinoma, medications that cause hyperprolactinemia, or organic diseases in the hypothalamic/pituitary region. Of these, drug-induced hyperprolactinemia is the most common. When prolactinoma is suspected, MRI imaging is warranted. Although microtumors are not usually obvious on simple MRI scans, gadolinium contrast-enhanced MRI can clearly detect them in most cases (Figure 1). In prolactinomas, tumor volume usually correlates with serum PRL levels. In the case of giant prolactinoma and moderately elevated PRL levels, the hook effect should be suspected. The hook effect is due to saturation of the assay antibodies, and diluted samples should be used for PRL measurement [22]. Regarding an endocrinological examination, the TRH loading test in prolactinomas often shows high basal PRL with low response. However, it is not specific and should be used as an ancillary test. The bromocriptine loading test is helpful for predicting the effectiveness of DA, as it suppresses PRL to more than half the basal value [1].

**Table 1 cancers-14-03604-t001:** Causes of hyperprolactinemia.

1. Pituitary disease. (1) Prolactinoma (2) Acromegaly (with simultaneous prolactin production)
2. Hypothalamic and pituitary stalk disease (1) Tumor (craniopharyngioma, Rathke’s cleft cyst, germ-cell, etc.) (2) Inflammation, Granulomatous disease (hypophysitis, salcoidosis, etc.) (3) Vascular disease (hemorrhage, infarction)
3. Medications (1) Dopamine antagonist (metoclopramide, etc.) (2) Anti-psychotic drug, Anti-depressant (chlorpromazine, haloperidol, etc.) (3) Anti-hypertensive drug (reserpine, verapamil, etc.) (4) H2 blocker (cimetidine, etc.) (5) Estrogens
4. Primary hypothyroidism
5. Macroprolactinemia
6. Others (1) Chronic renal failure (2) Neurogenic hyperprolactinemia (chest wall lesions: trauma, burns, eczema) (3) Ectopic prolactin-producing tumor (4) Idiopathic hyperprolactinemia

The table was created by modifying The Guide to Diagnosis and Treatment of Hypothalamic Pituitary Disease [23].

## 5. Medical Treatment

The goals of prolactinoma treatment are to counteract hypogonadism by suppressing hyperprolactinemia and decreasing the size of the tumor [1,2]. Pharmacotherapy is the first-line treatment for prolactinomas, and dopamine agonists (DAs) represent the primary therapy for almost all prolactinomas, including microtumors (less than 1 cm in diameter, macrotumors (greater than 1 cm), or giant tumors (greater than 4 cm). DAs, such as bromocriptine or cabergoline, are highly effective in suppressing PRL secretion and reducing tumor size (Figure 1). Long-acting DA (cabergoline) is often used in clinical practice because it is effective in small intermittent doses (0.25–0.5 mg once a week) without major side effects. Dosage reduction or discontinuation should be considered when normalization of serum PRL levels and disappearance of pituitary tumors are achieved after two or more years of DA treatment [1,24].

The normalization rate of serum PRL with bromocriptine was 78 and 72% for patients with microtumor and macrotumor, respectively [24]. Cabergoline is highly effective in normalizing serum PRL levels and reducing tumor size. In female patients treated with cabergoline, 83% achieved a normalized PRL; 72% had a recovered menstrual cycle, and only 3% discontinued because of adverse effects. Of those treated with bromocriptine, 59% achieved a normalized PRL; 52% had a recovered menstrual cycle, and 12% discontinued because of adverse effects [25]. In another study, cabergoline normalized PRL levels in 92% of patients with microprolactinoma and in 77% of patients with macroprolactinoma [26]. Cystic prolactinomas were thought to be resistant to drug therapy, but recently cabergoline has been shown to be beneficial [27]. Drug therapy for giant prolactinomas should be continued if they respond well. If the response to DAs is insufficient, surgical treatment should be considered to debulk the tumor volume, which may improve postoperative medical control.

DAs may produce side effects such as gastrointestinal symptoms, orthostatic hypotension, and nasal obstruction, as well as psychiatric symptoms, such as depression, anxiety, and insomnia [1,2]. The intravaginal administration of bromocriptine was effective in women and had diminished gastrointestinal side effects [28]. Furthermore, impulse disorders (e.g., pathological gambling, hypersexuality, and compulsive shopping or eating) have been described. The mechanism of impulse disorder is considered as DAs effect on the mesolimbic system via the subtype 3 dopamine receptor (D3 receptor). Discontinuing DAs usually reverses these side effects [29,30]. Long-term treatment with DAs may cause valvular heart disease. Although a large follow-up study did not support a clinical connection between DAs treatment and cardiac valvular disease, a meta-analysis that evaluated patients who received cabergoline treatment reported an increased risk of tricuspid regurgitation [31,32]. An echocardiogram should be conducted for patients with an audible murmur, those treated for more than 5 years with a dose of more than 3 mg per week, or those who maintained cabergoline treatment after age 50 [33]. In addition, when DAs are used for giant tumors invading the skull base, spinal fluid rhinorrhea and meningitis may occur as the tumor shrinks. In this case, patients are treated with antibiotics, or if necessary, the skull base is repaired by neurosurgery. Discontinuing DAs is not recommended because it may cause tumor relapse [34,35].

Prolactinomas are sometimes resistant to DA pharmacotherapy, and the estimated prevalence of DA resistance is 20–30% for bromocriptine and around 10% for cabergoline. In a study of 122 patients with macroprolactinoma, 80% of bromocriptine-resistant patients achieved normal PRL levels using cabergoline, and most patients resistant to standard doses of cabergoline responded to larger doses [24,36,37,38]. DA resistance was more prevalent in macroprolactinomas, invasive tumors, and male patients; 79% achieved normal PRL levels with a standard dose of cabergoline; 15% required higher doses; 6% were resistant. A T2-weighted MRI intensity of prolactinoma may help predict the response to DAs: the heterogeneity of tumoral T2 signal could be used as a predictive factor of DA resistance [39]. Regarding molecular mechanisms, DA resistance is associated with reduced subtype 2 dopamine receptor (D2 receptor) expression, particularly in the long-acting form of the D2 receptor [40,41]. In addition, factors downstream of the D2 receptor might contribute to DA inhibition, such as alterations in cytoskeleton protein filamin A or nerve growth factor (NGF) receptors [42,43]. Transforming growth factor beta (TGFβ) is also identified as a potential molecule for inducing DA resistance [44,45].

## 6. Surgical Treatment

The common indications for surgery in prolactinomas are resistance or intolerance to DA or the failure of the maximum dose of DA to lower the PRL or reduce tumor volume in macroprolactinomas. Acute complications (e.g., pituitary apoplexy, CSF leakage, or symptomatic tumor enlargement during pregnancy in DA-resistant cases) are rare indicators. Debulking surgery may be considered for women with macroprolactinomas who are planning to become pregnant. Other recently seen indications are patients with mainly cystic tumors or younger patients unwilling to undergo long-term DA treatment whose tumor is likely to be completely resected [2].

DA resistance is defined as the inability to achieve a normalization of serum PRL levels and a 50% reduction in tumor size at least 3–6 months after the weekly administration of the maximum-tolerated dose of a DA: 15 mg of bromocriptine or 1.5–3.0 mg of cabergoline [36]. If side effects make it difficult to increase the dose of a DA, a dose lower than this is the maximum. However, side effects of cabergoline were recorded in 68% of women, but only a few patients discontinued it [46].

Most surgeries for prolactinomas are performed by transsphenoidal surgery, and endoscopic transsphenoidal surgery is currently the mainstream method. A craniotomy is required depending on the location of the tumor. In a meta-analysis, remission rates were similar in patients who underwent endoscopic and microscopic surgery. It was reported that the neurosurgeon’s expertise, not the surgical approach, is more important for the surgical outcomes [2,47].

As for complications of transsphenoidal surgery, surgery-related mortality was 0%; persistent enuresis was 2%, meningitis was 1%, and the cerebrospinal fluid leak was 3%. In the meta-analysis, hypopituitarism was found in only 2%, hypoadrenocorticism in 1–2%, hypogonadism in 3–6%, and hypothyroidism in 1–6%. However, these risks are much greater in inexperienced centers [47]. Since the main purpose of prolactinoma surgery is to restore gonadal function, it becomes meaningless if it results in central hypogonadism.

A meta-analysis showed that the rate of gross tumor resection was inversely proportional to tumor size, with gross tumor resection achieved in 98–100% of microprolactinomas and 69–85% of macroprolactinomas. The overall postoperative and follow-up remission rates, having a median, mean follow-up period of 2.8 years using the random-effects model, were 0.62 and 0.61, respectively. The recurrence rate was 0.16 as assessed at a median follow-up period of 3.8 years [48]. The recurrence rate after surgery increased as the follow-up period increased [49]. However, in DA-resistant or intolerant patients, surgical tumor reduction may reduce, if not eliminate, the required DA dose or improve hormone control. Long-term remission has been reported to be highly controlled with a combination of surgery and a DA [46]. Postoperative remission was more likely to occur in microprolactinomas than in macroprolactinomas, and giant prolactinomas (>4 cm) rarely went into remission from surgery alone [50]. The success rate of surgery is highly dependent on the experience and skill of the neurosurgeon as well as the tumor size and degree of invasiveness. The remission rate was higher in centers with a large number of surgical cases than in centers with a small number [49]. Preoperative visual disturbances and headaches improved in the majority of patients after surgery [51].

The presence of cavernous sinus invasion and the initial serum PRL levels have often been reported as predictors of a surgical outcome. Even in intracellular prolactinomas, the remission rate was higher for midline-located tumors than those that are laterally located [52]. Similarly, prolactinomas with a lower Knosp grade have a higher remission rate [53]. Preoperative PRL levels <200 ng/mL were associated with a higher rate of long-term remission [52]. On the other hand, a preoperative PRL level >500 ng/mL is unlikely to result in a surgical cure, and >1000 ng/mL is unlikely to result in biochemical control [54]. A postoperative serum PRL level <10 ng/mL is associated with fewer recurrences [55,56].

In two meta-analyses of surgery versus DA, surgery resulted in 67–88% long-term remission after transsphenoidal surgery and 34–52% long-term remission after DA withdrawal, whereas the normalization of PRL with DA was achieved in 81% of cases [47,57]. Another meta-analysis reported that although there was no difference in the three-month results between the surgical and medical treatment, the remission rate was higher in the medical-treatment group at 12 months or longer [58]. Although many reports recommending surgery emphasize the side effects of DA treatment and the disadvantages of long-term use, it should be noted that the goal of treating prolactinomas, especially microprolactinomas, is to reverse gonadal dysfunction, not necessarily to wean the patient off of DA treatment.

## 7. Radiation Therapy

The aims of radiation therapy for prolactinomas are (1) to inhibit the growth of the tumor, (2) suppress hormone secretion, and (3) suppress the further progression of the tumor remnants that pathologically indicate aggressive behavior through adjuvant therapy. There are different varieties of radiotherapy. Fractionated external beam radiotherapy (EBRT), known as conventional radiotherapy, had been used mainly for pituitary tumors until recently. Now, stereotactic radiosurgery (SRS) or stereotactic radiotherapy (SRT) tends to be preferred because each allows the target zone to be irradiated while sparing the surrounding or peripheral healthy tissue. Hormone normalization with SRS takes less time [59]. However, EBRT has also improved as an intensity-modulated radiotherapy (IMRT) [60].

SRS is an effective option for controlling the growth of residual or recurrent tumors after surgical resection, but the rate of endocrine improvement or remission is low in functioning PitNETs. In meta-analyses, SRS resulted in tumor control rates of 86–100%, and endocrine remission rates varied between 6 and 81%, with remission rates increasing over time [61].

The most common side effect associated with SRS is hypopituitarism, with the incidence of new hormone deficiencies observed at 4.5–42%. The reported incidence of hypopituitarism with EBRT is approximately 50% 10 to 20 years after treatment. Since one of the purposes of prolactinoma treatment is to restore gonadal function, radiation therapy may be contra-indicated if it causes central hypogonadism. New neurological or visual defect rates ranged from 0 to 5% after SRS [61,62].

## 8. Pregnancy

Prolactinomas in young women often coincide with pregnancy and childbirth, and thus careful management is required. Although bromocriptine and cabergoline have been shown to have no adverse effects on the course of the pregnancy or the fetus, DAs should be discontinued in principle if pregnancy occurs during drug therapy [63,64]. There were no abnormalities in maternal–fetal outcomes in 6272 pregnancies with bromocriptine or in 1061 with cabergoline [65]. Regarding the risk of tumor growth during pregnancy, a systematic review indicated 2.4% symptomatic tumor growth in microprolactinomas, 21% in macroprolactinomas, and 4.7% in macroprolactinomas with previous surgery or radiotherapy [63]. Since there have been a few cases of tumor growth during the interruption of DAs during pregnancy, the course of clinical symptoms (e.g., headache or visual field disturbances) should be carefully monitored [1,63,66]. In healthy women, the high estrogen levels in pregnancy cause lactotroph hypertrophy and hyperprolactinemia, and serum PRL levels gradually elevate during the course of gestation; thus, serum PRL is not an indicator of prolactinoma progression. An MRI scan without gadolinium should be performed if a patient with macroprolactinoma has severe headaches or a visual field defect. DAs or transsphenoidal surgery should be considered if aggressive macroprolactinoma is evident [63,64].

## 9. Giant Prolactinoma

A giant prolactinoma is defined as prolactinoma with a tumor >4 cm and very high serum PRL levels. It is a rare type of PitNET accounting for 0.5–4.4% of PitNETs and 2–3% of prolactinomas. The giant prolactinoma occurs predominantly in males, with a male to female ratio of 9:1 [67,68]. Extremely high levels of PRL cause the “hook effect” [22].

As mentioned above, it has been reported that DA treatment for giant prolactinomas results in tumor shrinkage and an improved visual field: 90% showed tumor shrinkage alone, and PRL was also normalized in approximately 75–80% of cases with cabergoline treatment [67]. Surgery alone rarely cures giant prolactinomas. Most patients do not achieve hormonal remission after surgery, and surgical mortality and morbidity are high [68]. However, lifelong continuous DA therapy is required in almost all patients to maintain PRL suppression and prevent tumor regrowth [69].

Especially in giant prolactionomas, tumor shrinkage from DA treatment may rarely result in spontaneous CSF leakage. In such cases, bone defects of the cellular floor are seen on a CT scan due to tumor invasion [70].

## 10. Pituitary Carcinoma and Aggressive Prolactinoma

PitNETs are defined as malignant only when distant metastases are present and are called “pituitary carcinomas”. Malignant prolactinomas are reported to metastasize after a latency period of 2 months to 22 years, with an average of 4.7 years. The prognosis for pituitary carcinoma is poor, with an average survival of 10 months after diagnosis of metastasis. Moreover, the WHO classification defines an “aggressive PitNET” as having a radiologically invasive tumor and an unusually rapid tumor growth rate or clinically relevant tumor growth despite optimal standard therapies (surgery, radiotherapy, and conventional medical treatments). Although aggressive prolactinomas are essentially DA-resistant, prolactinomas that are hormonally uncontrolled but without tumor progression should not be considered aggressive PitNETs [71,72]. Prolactinomas are the second-most common tumor type treated with temozolomide after corticotroph pituitary tumors and represent about 30% of cases, and most DA-resistant prolactinomas are macrotumors. In a European Society of Endocrinology survey, 75% of aggressive PitNETs or pituitary carcinomas occurred in men [73,74].

Temozolomide is an alkylating agent that has shown efficacy in glioblastomas. Subsequently, temozolomide was reported to be effective against aggressive PitNETs and pituitary carcinomas (Figure 2). In 2017, the European Society of Endocrinology Clinical Practice Guidelines advised using temozolomide as a first-line chemotherapy treatment for aggressive PitNETs and carcinomas [74]. A recent systematic review estimated the five-year overall survival at 57% for all patients treated with temozolomide [73]. The response rate for prolactinomas and corticotroph tumors is about 50%, which compares favorably with nonfunctioning or GH-secreting tumors. After discontinuing temozolomide, the frequency of tumor regrowth ranged from 27 to 67%, with progression-free survival mainly in the range of 8 to 18 months. Even if pituitary tumors respond initially to temozolomide treatment, they often develop a resistance to it, so a second course of temozolomide treatment after regrowth is almost always ineffective [75].

O6-methylguanine-DNA methyltransferase (MGMT) is a DNA repair protein that removes the alkylation adducts induced by temozolomide and counteracts the antitumor actions of temozolomide. In addition, mutations in mismatch repair proteins, particularly MSH6, independent of MGMT, increase temozolomide resistance. These can be evaluated by immunohistochemistry, and a good response to temozolomide was reported when MGMT immunostaining was negative. Low MGMT expression is more common in aggressive PitNETs and pituitary carcinomas than in non-aggressive PitNETs, and prolactinomas are more likely to have low MGMT expression than other subtypes [75,76].

## 11. New Therapeutic Approach

Recently, the effectiveness of a new medical treatment using estrogen receptor (ER) antagonists such as tamoxifen or raloxifene was validated and showed promising results [77]. Considering the fact that ER is expressed in most of the surgical prolactinoma specimens examined and that, as described above, estrogen facilitates lactotroph tumor cell growth, the addition of an ER antagonist to conventional dopamine receptor agonist is reasonable and awaits further study. More recently, the effectiveness of the mammalian target of rapamycin (mTOR) inhibitor or tyrosine kinase inhibitor has also been reported [78,79].

## 12. Conclusions

Prolactinomas are a common disease among pituitary disorders, most of which can be treated by DAs; however, surgical treatment is required in some cases. Recently, the efficacy of temozolomide for malignant prolactinomas was reported.

## Figures and Tables

**Figure 1 cancers-14-03604-f001:**
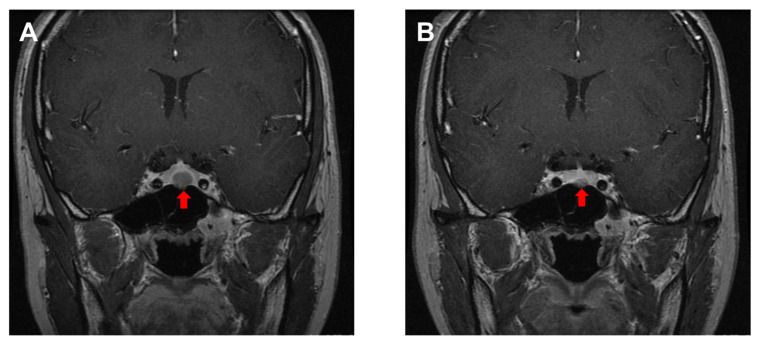
A representative case of enhanced T1-weighted MRI findings of prolactinoma. A 17-year-old woman presented with galactorrhea and amenorrhea. Serum prolactin (PRL) was 407 ng/mL, and gadolinium-enhanced MRI showed a pituitary mass lesion (arrow, **A**). The patient was diagnosed with prolactinoma, and cabergoline (0.25 mg per week) was prescribed and then increased (0.5 mg per week). Two years later, serum PRL was 15 ng/mL, and the MRI scan revealed that the pituitary microtumor had shrunk (arrow, **B**).

**Figure 2 cancers-14-03604-f002:**
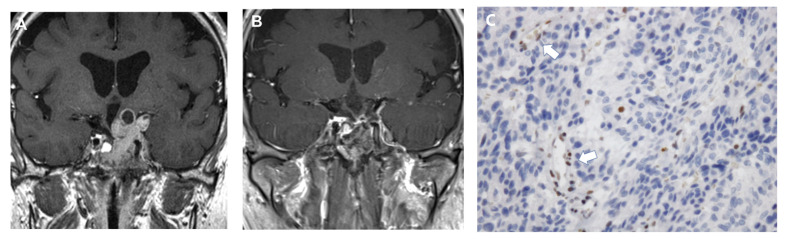
A case of refractory aggressive prolactinoma treated with temozolomide. (**A**): Coronal section of contrast-enhanced T1 weighted image. (**B**): Sagittal section. (**C**): MGMT immunostaining. Vascular endothelial cells show positive staining in the nucleus (arrow), while tumor cells are negative. This patient was a 64-year-old man. Although he underwent transsphenoidal surgery three times and was treated with 7 mg/week cabergoline, the tumor progressed. After commencing temozolomide treatment, the tumor shrank markedly, and serum prolactin decreased to below the reference range. Temozolomide was withdrawn after 34 courses, and no tumor recurred over the next 3 years.

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
