# Peer review of "Update in Pathogenesis, Diagnosis, and Therapy of Prolactinoma"

_cancers, 2022, doi:10.3390/cancers14153604_

Round 1
Reviewer 1 Report
This is a review article regarding prolactinoma, covering the pathogenesis, diagnosis, and management of prolactinomas.
This article is of interest to physicians involved in the care of patients with prolactinomas.
I would like to request some revisions to consider publication in "cancers".
Informed consent from the patients, who agreed on the use of their clinical data for figures 1 and 2, is required.
I also would like to ask the authors to consider mentioning the intravaginal administration of dopamine agonists for patients who develop GI symptoms from dopamine agonists, as it would be informative to endocrinologists who see patients with prolactinoma.
Sometimes, dopamine agonists are described as "Das". These should be "DAs".
I would be glad to review the revised manuscript in the near future.
Author Response
Reviewer #1
Informed consent from the patients, who agreed on the use of their clinical data for figures 1 and 2, is required.
Response:
We thank reviewer#1’s review and comments.
We used Figures 1 and 2 after obtaining informed consent from each patient at Kochi Medical School Hospital and Toranomon Hospital.
I also would like to ask the authors to consider mentioning the intravaginal administration of dopamine agonists for patients who develop GI symptoms from dopamine agonists, as it would be informative to endocrinologists who see patients with prolactinoma.
Response:
According to reviewer#1’s comment, we added a sentence as follows: “Intravaginal administration of bromocriptine was effective in women with diminished gastrointestinal side effects [28].” in line 220 – 221.
Sometimes, dopamine agonists are described as "Das". These should be "DAs".
Response:
According to reviewer#1’s comment, we corrected "Das" to "DAs" in lines 231 and 234.

Reviewer 2 Report
The authors performed a very interesting review on pathogenesis, epidemiology, and medical and invasive treatment of Prolactinomas. I think that such a review could be interesting mainly for physicians that usually do not encounter pituitary disease during their clinical activity.
I have two issues: one is related to the manuscript structure: this is a sort of narrative review, that in some part appear more similar to a book chapter than to an article. Authors should state the methodology for the review, whereas it does not fulfill the criteria for systematic review.
The other issue is related to the male patients with PRL tumors: I fully agree that the epidemiology of such tumors leads to describing mainly signs and symptoms in women and that in male patients often the clinical presentation is secondary to the mass effect rather than to endocrinological dysfunction. However, hypogonadism is the most common form of hypopituitarism in men with macroprolactinoma, and the entity of recovery after treatment is a relevant medical issue, mainly in younger patients. Authors should deeper describe this effect [i.e., see Al Dahmani et al. Proportion and predictors of Hypogonadism Recovery in Men with Macroprolactinomas treated with dopamine agonists. Pituitary. 2022 Jul 6. doi: 10.1007/s11102-022-01242-y. PMID: 35793046.]
Author Response
Reviewer #2
I have two issues: one is related to the manuscript structure: this is a sort of narrative review, that in some part appear more similar to a book chapter than to an article. Authors should state the methodology for the review, whereas it does not fulfill the criteria for systematic review.
Response:
We thank reviewer#2’s review and comments.
Editorial office invited us to make “review” on “Special issue: Pituitary tumors: molecular insights, diagnosis, and targeted therapy” in Cancers. There were no instructions for “systematic review”, therefore we performed “narrative review” on prolactinoma.
According to reviewer#2’s comment, we added “Search strategies” in lines 426-429 describing the selection method of references.
The other issue is related to the male patients with PRL tumors: I fully agree that the epidemiology of such tumors leads to describing mainly signs and symptoms in women and that in male patients often the clinical presentation is secondary to the mass effect rather than to endocrinological dysfunction. However, hypogonadism is the most common form of hypopituitarism in men with macroprolactinoma, and the entity of recovery after treatment is a relevant medical issue, mainly in younger patients. Authors should deeper describe this effect [i.e., see Al Dahmani et al. Proportion and predictors of Hypogonadism Recovery in Men with Macroprolactinomas treated with dopamine agonists. Pituitary. 2022 Jul 6. doi: 10.1007/s11102-022-01242-y. PMID: 35793046.]
Response:
According to reviewer#2’s comment, we added a sentence as follows: “In addition, hypogonadism is a common form of hypopituitarism in men with macroprolactinoma, and the entity of hypogonadism recovery after treatment is a relevant medical issue, mainly in younger patients [15].” in lines 132 – 134.

Round 2
Reviewer 1 Report
The manuscript has been revised with the authors' comments. I understood the authors obtained informed consent from the patients for Figures 1 and 2. However, I still would like to request the authors describe the information regarding the informed consent in the manuscript. With the documentation of the informed consent in the manuscript (Informed Consent Statement, Line 432), I would recommend this manuscript for publication in Cancers. This review article will be very informative to healthcare professionals involved in the care of patients with prolactinomas.
Author Response
We thank Reviewer#1’s review and comments. We have added informed consent statement in Line 432.

Reviewer 2 Report
The manuscript is now suitable for publication
Author Response
We sincerely thank Reviewer#2's review.